# HSF1 Can Prevent Inflammation following Heat Shock by Inhibiting the Excessive Activation of the *ATF3* and *JUN*&*FOS* Genes

**DOI:** 10.3390/cells11162510

**Published:** 2022-08-12

**Authors:** Patryk Janus, Paweł Kuś, Natalia Vydra, Agnieszka Toma-Jonik, Tomasz Stokowy, Katarzyna Mrowiec, Bartosz Wojtaś, Bartłomiej Gielniewski, Wiesława Widłak

**Affiliations:** 1Maria Sklodowska-Curie National Research Institute of Oncology, Gliwice Branch, Wybrzeże Armii Krajowej 15, 44-102 Gliwice, Poland; 2Department of Systems Biology and Engineering, Silesian University of Technology, Akademicka 16, 44-100 Gliwice, Poland; 3Department of Clinical Science, University of Bergen, Postboks 7800, 5020 Bergen, Norway; 4Laboratory of Sequencing, Nencki Institute of Experimental Biology, Polish Academy of Sciences, 3 Pasteur Street, 02-093 Warsaw, Poland; 5Laboratory of Molecular Neurobiology, Nencki Institute of Experimental Biology, Polish Academy of Sciences, 3 Pasteur Street, 02-093 Warsaw, Poland

**Keywords:** cancer biology, gene expression regulation, heat shock response, HSF1, inflammation

## Abstract

Heat Shock Factor 1 (HSF1), a transcription factor frequently overexpressed in cancer, is activated by proteotoxic agents and participates in the regulation of cellular stress response. To investigate how HSF1 level affects the response to proteotoxic stress, we integrated data from functional genomics analyses performed in MCF7 breast adenocarcinoma cells. Although the general transcriptional response to heat shock was impaired due to HSF1 deficiency (mainly chaperone expression was inhibited), a set of genes was identified, including *ATF3* and certain *FOS* and *JUN* family members, whose stress-induced activation was stronger and persisted longer than in cells with normal HSF1 levels. These genes were direct HSF1 targets, suggesting a dual (activatory/suppressory) role for HSF1. Moreover, we found that heat shock-induced inflammatory response could be stronger in HSF1-deficient cells. Analyses of The Cancer Genome Atlas data indicated that higher *ATF3*, *FOS*, and *FOSB* expression levels correlated with low HSF1 levels in estrogen receptor-positive breast cancer, reflecting higher heat shock-induced expression of these genes in HSF1-deficient MCF7 cells observed in vitro. However, differences between the analyzed cancer types were noted in the regulation of HSF1-dependent genes, indicating the presence of cell-type-specific mechanisms. Nevertheless, our data indicate the existence of the heat shock-induced network of transcription factors (associated with the activation of TNFα signaling) which includes HSF1. Independent of its chaperone-mediated cytoprotective function, HSF1 may be involved in the regulation of this network but prevents its overactivation in some cells during stress.

## 1. Introduction

The heat shock response (HSR) is defined as an inducible molecular response to a disruption of protein homeostasis which results in the elevated expression of cytoprotective genes to protect the proteome against toxic insults (such as increased temperatures, oxidative stress, heavy metals, etc). Such protection is primarily provided by the evolutionarily conserved heat shock proteins (HSP) and other molecular chaperones, which help to renature proteins unfolded during stress, or direct them for degradation when repairing is impossible [1]. Moreover, HSPs prevent or suppress apoptosis by modulating both the mitochondrial- or death receptor-mediated apoptotic pathways and by interfering with caspase activation at several different levels [2]. Many chaperones’ encoding genes are directly activated by the Heat Shock Factor 1 (HSF1). HSF1 is constitutively expressed in most tissues and cell types, where it is kept inactive in the absence of stress stimuli. It becomes activated under stress by forming trimers, which, in turn, bind specifically to Heat Shock Elements (HSEs) throughout the genome. The HSE consensus sequence is a tandem array of at least three oppositely oriented “nGAAn” motifs or a degenerate version thereof. HSF1-activated genes can initiate further signaling events (secondary and tertiary), creating a branching network that in most cells allows the mounting of a multifaceted response that promotes survival and prepares the cell for additional insults [3]. Consequently, HSF1-deficient cells exhibit increased heat sensitivity and cannot develop thermotolerance [4]. HSF1 also plays important role in non-stress regulation, such as development and metabolism [5]. Dysregulation of HSF1 and its target genes are associated with the disease. For example, HSF1 is often overexpressed in cancer cells where it supports a malignant phenotype [6,7,8] whereas HSF1 hypoactivation in neurodegenerative disorders results in the formation of toxic aggregates [9]. There are exceptions to cytoprotective signaling during cellular stress and HSF1 can activate pro-death signaling in some cells through induction of the pro-apoptotic *PMAIP1* gene [10]. Simultaneously, HSPs activation is blocked in such cells [11]. Therefore, understanding the regulation of HSF1 and the specificity of its action could open up new therapeutic opportunities [12].

Genome-wide analyses provided evidence that HSF1 does not directly regulate the induction of every transcript that accumulates after heat shock in mammalian cells [13]. A combination of chromatin immunoprecipitation microarray analysis and time-course gene expression microarray analysis with and without siRNA-mediated inhibition of HSF1 enabled the identification of genes regulated directly and indirectly by HSF1 [14]. Later, more advanced studies combined ChIP-seq and PRO-seq data from heat-shocked wild-type and *Hsf1*^−/−^ mouse embryonic fibroblasts to demonstrate that the promoter-bound HSF1 is not responsible for the induction or repression of the majority of HS-regulated genes. HSF1 controls only a fraction of heat shock-induced genes and does so by increasing RNA polymerase II release from promoter-proximal pause [15]. In contrast to these results, however, there are also data suggesting that a large majority of heat-induced genes are positively regulated by HSF1 [16]. Herein, we performed a functional genomics study to analyze comprehensively the consequences of the HSF1 deficiency on the signaling pathways induced by heat shock in cancer cells.

## 2. Materials and Methods

### 2.1. Cell Lines and Treatments

Human MCF7 ERα-positive breast cancer cell line was purchased from the American Type Culture Collection (ATCC, Manassas, VA, USA). Human HAP1 parental control cell line (near-haploid cell line derived from the chronic myelogenous leukemia KBM-7) and HSF1 26bp deletion knockout cell line (HZGHC004801c008) were purchased from Horizon Discovery Ltd. (Cambridge, United Kingdom). Cells were cultured in DMEM/F12 medium (Merck KGaA, Darmstadt, Germany) supplemented with 10% fetal bovine serum (FBS) (EURx, Gdansk, Poland). HAP1 cells were cultured in IMDM with 10% FBS and penicillin-streptomycin (Merck KGaA). Cells were routinely tested for mycoplasma contamination. MCF7 cells with downregulated HSF1 using shRNA (stably transduced with lentiviruses) or HSF1 CRISPR/Cas9-mediated functional knockout (two individual clones) and corresponding control cell variants were obtained as described in detail previously [17] (collectively referred to herein as HSF1^def^ cells). For validation experiments, MCF7 and RKO (colon carcinoma) knockout cells created using a DNA-free CRISPR/Cas9 system described previously were used [10,17]. Six individual unaffected clones (HSF1+) or with the HSF1 functional knockout (HSF1−) were pooled each time before analysis. For heat shock, logarithmically growing cells were placed in a water bath at a temperature of 43 °C and allowed to recover for the indicated time in a CO_2_ incubator at 37 °C. The growth media were not replaced either before or after treatments.

### 2.2. Global Gene Expression Profiling and Analysis

Total RNA was isolated and processed from HSF1-proficient (SCR, MIX, WT) and HSF1-deficient (shHSF1, KO#1, KO#2) MCF7 cell variants, untreated, and two hours after one-hour heat shock at 43 °C, as described in [17]. In shHSF1 cells, the level of HSF1 was stably decreased by 90% by HSF1-specific shRNA (Appendix A). In KO#1 and KO#2 clones, the functional knockout was obtained using the CRISPR/Cas9 method. HSF1-proficient cell line variants included corresponding controls and wild-type cells. RNA-seq was performed in parallel with an experiment in another our project in which these cell variants were treated with estrogen (E2), thus cells were grown in phenol red-free media and dextran-activated charcoal-stripped FBS (PAN-Biotech GmbH, Aidenbach, Germany). Counts data for 18 RNA-seq libraries (data deposited in the NCBI GEO database; acc. no. GSE159802) were loaded into R (v. 4.0.3). RNA-seq libraries from E2-stimulated cells were kept in the analysis to improve the dispersion estimates, although their analysis was not a part of this research and is not included in this paper. Counts matrix was filtered using *filterByExpr()* function from edgeR package (v. 3.32.1, samples grouped by stimulation) [18], then differentially expressed genes were detected with DESeq2 (v. 1.30, *design = ~HSF1_condition + treatment*) [19]. Finally, the p-values were corrected for multiple testing using the Benjamini and Hochberg method. Volcano plots were plotted using the custom R function available in the github repository https://github.com/pawel125/omicsTools (accessed on 3 January 2022). For gene set enrichment analysis, we selected Hallmark, BioCarta, Reactome, and PID genesets from MSigDB (v. 7.2) [20] and merged it with the list of pathways downloaded from KEGG. Genes were ordered according to their p-value and tested for enrichment using the CERNO test [21] from the tmod package (v. 0.46.2) [22]. The most significant results of the gene set enrichment analysis were presented using ggplot2 package (v. 3.3.5) [23] and/or *tmodPanelPlot()* function from tmod. Upstream regulators were predicted using the ChIP-X Enrichment Analysis Version 3 (ChEA3) [24] based on the ReMap transcriptional regulators library constructed from human data.

### 2.3. RNA Isolation, cDNA Synthesis, and RT-qPCR

Total RNA was isolated using the Direct-ZolTM RNA MiniPrep Kit (Zymo Research, Irvine, CA, USA), digested with DNase I (Worthington Biochemical Corporation, Lakewood, NJ, USA), and cleaned with RNAClean XP beads (Beckman Coulter Life Science, Indianapolis, IN, USA). RNA (1 μg) was converted into cDNA as described [25]. Quantitative PCR was performed using a BioRad C1000 TouchTM thermocycler connected to the head CFX-96 (Bio-Rad Laboratories, Inc, Hercules, CA, USA). Each reaction was performed at least in triplicates using PCR Master Mix SYBRGreen (A&A Biotechnology, Gdynia, Poland). Expression levels were normalized against *GAPDH*, *ACTB*, *HNRNPK*, and *HPRT1*. The set of delta-Cq replicates (Cq values for each sample normalized against the geometric mean of four reference genes) for control and tested samples were used for statistical tests and estimation of the p-value. Shown are median, maximum, and minimum values of a fold-change vs. untreated control in HSF1+ cells. The primers used in these assays are described in Appendix A.

### 2.4. Protein Extraction and Western Blotting

Whole-cell extracts were prepared using RIPA buffer supplemented with Complete^TM^ protease inhibitors cocktail (Roche, Indianapolis, IN, USA) and phosphatase inhibitors PhosStopTM (Roche). Proteins (20–30 μg) were separated on 10% SDS-PAGE gels and blotted to a 0.45 μm pore nitrocellulose filter (GE Healthcare, Europe GmbH, Freiburg, Germany) using the Trans-Blot Turbo system (Thermo Scientific™ Pierce™ G2 Fast Blotter) for 10 min. Primary antibodies against HSF1 (1:2000; ADI-SPA-901) and HSP70 (1:5000, ADI-SPA-810), both from Enzo Life Sciences (Farmingdale, NY, USA); ATF3 (1:1000; #33593S, Cell Signaling Technology, Danvers, MA, USA); EGR3 (1:1000, #sc-390967) and HSPA8/HSC70 (1:5000, #sc-7298), both from Santa Cruz Biotechnology (Dallas, TX, USA); HSF2 (1:3000; #AF5227, R&D Systems, Minneapolis, MN, USA); HSF1 phospho S326 (1:3000; # ab76076, Abcam, Cambridge, UK); and ACTB (1:25,000, #A3854, Merck KGaA) were used. The primary antibody was detected by an appropriate secondary antibody conjugated with horseradish peroxidase (Thermo Fisher Scientific, Waltham, MA, USA) and visualized by an ECL kit (Thermo Fisher Scientific) or WesternBright Sirius kits (Advansta, Menlo Park, CA, USA). Imaging was performed on X-ray film. The experiments were repeated at least twice and blots were subjected to densitometric analyses using Image Studio Lite v. 5.2.5 software to calculate relative protein expression after normalization with loading controls (statistical significance of differences was calculated using a *T*-test).

### 2.5. Chromatin Immunoprecipitation (ChIP), Global Profiling of Chromatin Binding Sites, and ChIP-qPCR

The ChIP assay in MCF7 wild-type cells, untreated and heat-shocked at 43 °C for 15 min, was performed according to the protocol from the iDeal ChIP-seq Kit for Transcription Factors (Diagenode, Denville, NJ, USA) using anti-HSF1 antibody (#ADI-SPA-901, Enzo), sequenced, and analyzed as described in detail in [26]. ChIP-Seq heatmaps were prepared using the peakHeatmap function from the ChIPseeker Bioconductor package (v. 1.26.2), with margins of 3000 nucleotides upstream and downstream from the promoter. The raw ChIP-seq data were deposited in the NCBI GEO database; acc. no. GSE137558 (GSM4081758, GSM4081759, and GSM4081762). For ChIP-qPCR (described in detail in [26]) in HSF1+ and HSF1− MCF7 cells, anti-HSF2 antibody (#AF5227, R&D Systems) was used in addition to anti-HSF1. The sequences of used primers are presented in Appendix A.

### 2.6. ChIP-Seq and RNA-Seq Data Integration

To identify genes directly regulated by HSF1 in response to heat shock in MCF7 cells, a BETA (Binding and Expression Target Analysis) BASIC package v. 1.0.7 [27] located on the Cistrome server [28] was implemented. HSF1 ChIP-seq differential peaks (HS-treated versus untreated cells: 18,645 peaks with the number of tags ≥40 and peak score ≥100 in standard bed file format) were integrated with differential expression data from RNA-seq (HS-treated versus untreated HSF1+ cells: a set of unique 13,970 gene Ensembl IDs in the BETA specific file format containing gene symbol, regulatory status (value with + or −) and statistical value (padj)). Hg19 was used as a reference genome and additionally, we included the CTCF (CCCTC-Binding Factor) boundary to filter peaks around a gene.

### 2.7. Proximity Ligation Assay

To detect the CTCF interactions with HSF1, the DuoLink in situ Proximity Ligation Assay (PLA) (Merck KGaA) was used according to the manufacturer’s protocol. HSF1+ and HSF1− MCF7 cells were plated onto Nunc^®^ Lab-Tek^®^ II chambered coverglass (#155383, Nalge Nunc International, Rochester, NY, USA) one day before the experiment. Untreated and heat-shocked cells (15 min at 43 °C) were fixed for 15 min with 4% PFA solution in PBS, washed in PBS, and treated with 0.1% Triton-X100 in PBS for 5 min. After washing, slides were incubated in a blocking solution and immunolabeled (overnight, 4 °C) with primary antibodies diluted in the DuoLink^®^ Antibody Diluent: rabbit anti-CTCF (1:600; #C15410210, Diagenode) and mouse anti-HSF1 (1:200; #sc-17757, Santa Cruz Biotechnology). Then, the secondary antibodies with attached PLA probes were used. Signals of analyzed complexes were observed using Carl Zeiss LSM 710 confocal microscope with ZEN navigation software (Carl Zeiss AG, Oberkochen, Germany); red fluorescence signal indicated proximity (<40 nm) of proteins recognized by both antibodies [29]. Z-stack images (12 slices; 5.5 μm) were taken at ×630 magnification. Spots identified in nuclei were counted manually using the ImageJ software from at least 100 cells for each condition. Outliers were determined using the Tukey criterion. For each dataset, the normality of distribution was assessed by the Shapiro–Wilk test. In the case of non-Gaussian distribution, the Kruskal–Wallis ANOVA was applied for the verification of the hypothesis on the equality of medians with the Dunn test for pairwise comparisons with Bonferroni and Benjamini–Hochberg correction. *p* = 0.05 was selected as a statistical significance threshold.

### 2.8. Measurement of Secreted Cytokines by ELISA

HSF1+ and HSF1− MCF7, RKO, and HAP1 cells were seeded on 6-well plates in 2 mL of medium. The next day, when the cell culture reached 50% confluence, the medium was changed. On the third day, cells were heat-shocked at 43 °C in a water bath for one hour and recovered for 3 h, 6 h, and 24 h. The media were collected, centrifuged (2 min, 2000 rpm), aliquoted on an ice bath, and then frozen at −80 °C. Cells remaining on the plates were trypsinized and the total number of cells (viable and dead) was estimated using a Bürker chamber. Control cells, not subjected to heat shock, were harvested along with those harvested after 24 h of recovery. The experiment was performed in triplicate. For ELISA, the collected media were thawed in an ice bath. The concentration of TNF alpha (Cat.No.: 88-7346), IL-6 (Cat.No.: 88-7066), IL10 (Cat.No.: 88-7106), IL1 beta (Cat.No.: 88-7261), IL2 (Cat.No.: 88-7025), and IL4 (Cat.No.: 88-7046), all from Invitrogen/Thermo Fisher Scientific, was determined according to the manufacturer’s instructions. All incubation steps were performed with shaking (200 rpm on a rotary shaker). Measurements were performed on the SPARK spectrophotometer (Cat.No.:1902008176; Tecan, Männedorf, Switzerland) at 450 nm and 570 nm. The values of 570 nm were subtracted from those of 450 nm and the cytokine concentration (in pg/mL) calculated from the standard curve was normalized to 1 million cells. The normality of the distribution was checked with the Shapiro–Wilk test. The analysis of the homogeneity of variance was performed with the Fisher–Snedecor test. For analysis of differences between compared groups with normal distribution, the appropriate Student’s *t*-test was performed. The analysis of the mean comparisons for the treated versus control samples was performed using the ANOVA test and Dunnett’s post hoc test after checking the assumption of homogeneity of the variance. For the ANOVA test homogeneity of variances was verified by Levene’s test. In the case of non-Gaussian distribution, the non-parametric Mann–Whitney U test was used for mean comparisons. *p* = 0.05 was selected as a statistical significance threshold.

### 2.9. TCGA (The Cancer Genome Atlas) Data Analysis: Data Retrieval, Selection of Cases, and Differential Expression Analysis

TCGA data analysis was performed using R v. 4.1.3. Clinical and RNA-seq (STAR-Counts) data from three TCGA projects (TCGA-BRCA, 1219 total samples; TCGA-COAD, 519 total samples; and TCGA-LAML, 150 total samples) were downloaded and prepared using the TCGAbiolinks package (v. 2.23.8) [30]. An additional file with clinical data containing estrogen receptor (ER) status, ‘nationwidechildrens.org_clinical_patient_brca.txt’, was downloaded directly from the GDC repository https://portal.gdc.cancer.gov (accessed on 23 March 2020). Finally, TCGA-BRCA patients were divided into ER+ and ER− groups based on the ER status and PAM50 from the clinical data. The ER-negative group was composed of the HER2 and Basal-like PAM50 subtypes, while the ER-positive group was composed of the Normal and Luminal PAM50 subtypes. For each tumor type, we defined two groups of patients in the following manner: (1) counts were normalized using the *vst()* function from the DESeq2 package (v. 1.34); (2) cases with extremely high/low HSF1 expression (below 1st quartile—1.5 IQR or above 3rd quartile + 1.5 IQR, IQR—interquartile range) were marked as outliers and excluded; (3) remaining cases were divided into three HSF1-level-based groups and the intermediate groups were excluded. For differential expression analysis, we used the *filterByExpr()* function from the edgeR package to remove all lowly expressed genes but those that we found differentially expressed in our RNA-seq analysis (1300 genes), which resulted in around 16,000–20,500 genes for tumor kept for statistical analysis. Then, counts matrices for each tumor were separately processed with *DESeq()* function from the DESeq2 package (*design = ~HSF1_group*) to identify differentially expressed genes between the groups of patients with high and low HSF1 levels. Finally, the *p*-values were corrected for multiple testing using the Benjamini and Hochberg method. Volcano plots were plotted using the VolcaNoseR web app [31]. Log2 fold change between −0.2 and 0.2 and padj >0.05 was considered as not significant. Correlation analysis was performed with the cor.test() function in R using Pearson’s method.

## 3. Results

### 3.1. HSF1 Deficiency Generally Impairs Transcriptional Response to Heat Shock but Results in Enhanced Induction of a Subset of Genes

To investigate how cancer cells deal with cellular stress in the case of HSF1 deficiency, we analyzed the global gene expression profiles in breast adenocarcinoma MCF7 cells with decreased HSF1 levels (HSF1-deficient, HSF1^def^) and corresponding control cells with normal HSF1 levels (HSF1-proficient, HSF1^prof^) (cell model described previously in [17]). HSF1 silencing or functional knockout (Appendix A) inhibited activation of *HSP* genes after heat shock, although full inhibition was only seen in some genes and in the complete absence of HSF1 (Appendix A). RNA-seq analysis (Appendix A) revealed that only a few genes were differentially expressed in untreated (Ctr) HSF1^prof^ and HSF1^def^ cells (including HSF1; Appendix A). In response to heat shock (HS), 1101 genes changed expression at least 2-fold in HSF1^prof^ cells (736 up and 365 down) (Figure 1A). As expected, these were primarily the well-known targets of HSF1 (i.e., *HSP* genes), thus terms related to the HSF1-mediated heat shock response (M27250, M27252, M27253, M27254, M27255), and to a lesser extent, other stress-related pathways were identified as overrepresented in the gene set enrichment analysis, GSEA (Figure 1D,E). All these terms were not enriched in heat-shocked HSF1^def^ cells. Generally, the number of genes identified as responding to heat shock in HSF1^def^ cells was smaller by over 30% compared to HSF1^prof^ cells (729 genes changed expression at least 2-fold: 451 up and 278 down) (Figure 1B), however, several pathways were more enriched in HSF1^def^ cells (e.g., M14339, M29832; Figure 1D and Appendix A). A direct comparison of the response to heat shock in both cell variants revealed that 38% of genes responding in HSF1^prof^ cells responded similarly in HSF1^def^ cells (change at least 2-fold, padj < 0.05; ~28% upregulated and ~10% downregulated; Figure 1F). Further analysis identified quantitative differences in the response between both cell variants. The loss of HSF1 resulted in weaker upregulation (or stronger inhibition) of 235 genes (Figure 1C) as expected. However, despite the generally weaker HS response in HSF1^def^ cells, 77 genes revealed a higher heat shock-affected expression in this cell variant (Figure 1C), either stronger induction or weaker inhibition. GSEA indicates that a large fraction of genes more strongly activated in HSF1^def^ cells was involved in TNFα signaling via NFκB (M5890) or the establishment of sister chromatid cohesion (M27177) (Figure 1D,E and Appendix A).

To validate the RNA-seq results, we selected 12 genes from the list of 77 genes that showed stronger heat-inducibility in HSF1^def^ cells (or weaker inhibition by heat shock; Figure 1C), and 7 additional genes (*CLCF1*, *CHAC1*, *EGR1*, *EGR3*, *FOSB*, *ID2*, *SMAD7*) that were also more strongly induced in HSF1^def^ cells, but did not meet preselected criteria (log2 fold change > 1.0 and padj < 0.05). Validation experiments were performed on three human cell lines: MCF7, HAP1, and RKO with functional HSF1 knockout and control lines with normal HSF1 levels (HSF1− and HSF1+, respectively). Analyses of HSF1 levels and *HSPA1* expression, a canonical target of HSF1, confirmed functional knockout of HSF1 in all three cell models (Figure 2A,B). Interestingly, the level of HSF1 in HAP1 cells was much lower than in MCF7 and RKO cells (Figure 2B) which may be related to the fact that HAP1 cells are near-haploid. RT-qPCR analyses of the response to heat shock (up to 6 h of recovery) revealed that all selected genes were more strongly induced in HSF1− MCF7 cells, which was consistent with the RNA-seq data (although a different cell model was used) (Figure 2A and Appendix A). Moreover, the majority of selected genes (*EGR1*, *EGR3*, *JUNB*, *KLF10*, *SMAD7*, *DUSP10*, *ID2*, *IER2*, *LINC00324*, *RND3*, *TRIB1*) showed stronger heat-induced activation also in HSF1− RKO and HAP1 cells. Some differences between cell lines were observed in the case of *ATF3*, *CHAC1*, *CLCF1*, *DDIT3*, *FOSB*, *TIPARP*, *VEGFA*, and *WEE1*, indicating that their regulation after heat shock can be cell-type specific. For validation at the protein level, we have chosen ATF3 (which in HAP1 cells showed a weaker transcriptional response in HSF1− than in HSF1+ cells) and EGR3 (since we previously found its dependence on HSF1 in response to estrogen treatment; [17]). Western blot analyses up to 24 h of recovery from heat shock confirmed the higher levels or delayed upregulation of EGR3 and ATF3 in stressed HSF1− cells (for details see Figure 2B).

### 3.2. HSF1 Has an Activating Function in the Heat-Induced Transcription but May Inhibit the Overactivation of Certain Genes including Those Coding for Transcription Regulators

ChIP-seq analysis (Appendix A) revealed a strong enrichment in HSF1 binding induced by heat shock in MCF7 cells: approximately 14 times more peaks were identified than in untreated cells and most of them were located in intergenic and intronic regions (Figure 3A). This is in line with a previous report that showed an involvement HSF1 in the induction of RNA-producing enhancers across the whole genome, but also the recruitment of HSF1 in CTCF-rich, non-transcribed chromatin [32], suggesting that interactions between HSF1 and CTCF are possible. Indeed, such interactions were later documented [33]. We also showed that HSF1 is likely to interact with CTCF in MCF7 cells (as assessed by the proximity ligation assay) and the number of interactions increased significantly after heat shock treatment (Figure 3B). Thus, remodeling of the genome-wide binding of HSF1 after heat shock could be mediated by CTCF. Interestingly, HSF1 is also likely to regulate the *CTCF* expression as it may bind to *CTCF* regulatory sequences following heat shock (Appendix A). However, we did not find any changes in *CTCF* transcription (RNA-seq data) and although CTCF protein levels may increase after heat shock (especially C-terminally truncated 70 kDa form which is likely to be a product of premature termination of translation [34]), this seems to be HSF1-independent (Appendix A). Consequently, in ChIP-seq (HS versus Ctr peaks) and RNA-seq (HS versus Ctr in HSF1^prof^ cells) data integration using the BETA software package [27], we included the CTCF boundary to filter peaks around a gene. This analysis revealed that HSF1 has rather an activating function, not repressive (Figure 3C; Appendix A). Among 736 genes upregulated at least 2-fold after heat shock in HSF1^prof^ cells (Figure 1A), 497 were predicted by the BETA tool to be regulated by HSF1, yet a closer examination (in IGV) of the remaining heat-induced genes revealed that HSF1 was additionally bound in regulatory regions of at least 36 of them. Thus, finally, about 80% of genes upregulated after heat shock (533 genes) might be directly bound by HSF1 (Appendix A, “HSF1-bound, HSF1^prof^_HS_up 2x” sheet). Interestingly, however, only part of them showed a significantly smaller fold change in heat-shocked HSF1^def^ cells (179 genes, padj of the difference <0.05; Appendix A, “FC_HSF1^prof^ > FC_HSF1^def^“ sheet), which confirms the dependence of their activation on HSF1 (examples are shown in Figure 3D,E). On the other hand, approximately 100 genes bound by HSF1 and activated after heat shock in HSF1^prof^ cells were still heat-activated in HSF1^def^ cells (Appendix A, “FC_HSF1^def^ > FC_HSF1^prof^” sheet). The list of such genes was then restricted to approximately 40 genes (based on HSF1 binding visible in IGV) (Appendix A, “HSF1^def^_not inhibited_selected” sheet). Table 1 shows the top 14 genes upregulated after heat shock with potential HSF1 binding but induced more strongly in the absence of HSF1. Notably, 9 out of 14 genes code for regulators of gene expression. These results suggest that HSF1 may not only have an activating function after heat shock but may also inhibit the transcription of certain genes (in particular expression regulators, which would otherwise excessively accelerate transcriptional response to heat shock).

### 3.3. Heat Shock-Induced Inflammatory Response Could Be Stronger in HSF1-Deficient Cells

Gene set enrichment analysis of RNA-seq results indicated that HSF1-proficient and HSF1-deficient MCF7 cells may differ in the heat shock-induced TNFα signaling via NFκB (Figure 1D,E). Moreover, some genes involved in the regulation of NFκB signaling are direct targets of HSF1 and are activated after heat shock only in HSF1+ cells (e.g., *NKRF*, *TRAF2*, see Figure 3D,E; also *TRAF3*, *NFKBID*, not shown). This prompted us to analyze the level of cytokines produced and secreted to culture media after heat shock by HSF1+ and HSF1− MCF7, RKO, and HAP1 cells. IL10, IL1β, IL2, and IL4 were not released under any condition by all analyzed cell variants (not shown) while *IL6* transcript was not detected in RKO cells. The ELISA assay combined with the RT-qPCR analyses (Figure 4) indicated that the heat shock-induced production of TNFα was higher in HSF1− cells. On the other hand, expression of *IL6* after heat shock was delayed in HSF1− MCF7 and HAP1 cells and started to rise as the decline in HSF1+ began. These results suggest that the inflammatory response induced by heat shock could be stronger and extended in cells with low levels of HSF1.

### 3.4. Transcriptional Regulatory Network Created during Heat Shock May Be Repressed by HSF1

HSF1 is a primary transcription factor induced by heat shock. Using the ChEA3 transcription factor enrichment analysis [24], it was predicted that HSF1 was responsible for the activation of ~14% of the genes (although our RNA-seq and ChIP-seq data integration analysis indicates that it may be a much larger fraction). In addition, several other transcription factors were predicted as upstream regulators of the heat shock-activated genes and they are mostly the same in both cell variants (Figure 5A). These included, but were not limited to, the various FOS and JUN subunits of the AP1 transcription factor and ATF3, which showed higher stress-induced expression in HSF1-deficient cells. It is noteworthy that out of 84 such genes (77 from Figure 1C and an additional 7 included in the RT-qPCR validation), 81 were protein-coding genes, and 47 of them were classified as regulators of gene expression (Appendix A). An analysis of potential protein–protein interactions using the STRING tool (version 11.5) [35] showed that the largest network is formed around FOS and JUN family members and ATF3 (the network contains also EGR1, EGR3, and other transcription factors; Appendix A). Moreover, we detected HSF1 binding in regulatory regions of *ATF3*, *JUN*, *JUND* (but not *JUNB*), *FOS*, *FOSL1*, *FOSL2*, and *FOSB* (Figure 5B). Thus, only *JUNB* seems to be induced by heat shock independently of HSF1 binding, while *FOSL2* was not upregulated (at least 2 h after HS) despite HSF1 binding (Figure 5B,C, see also Figure 2A and Figure 5E). HSF1 binding connected with upregulation of the expression of *ATF3*, *JUN* and *FOS* family members (as well as other transcriptional regulators; see Table 1 and Appendix A) after heat shock implicates that HSF1 may be involved in the creation of the stress-activated regulatory network. On the other hand, however, the lack of significant inhibition of the hypothetical components of this network (in the case of *JUN*, *JUND*, *FOSL1*) or even stronger activation (in the case of *ATF3*, *FOS*, *FOSB*) in HSF1-deficient cells suggests that HSF1 is not an obligatory component of this network and could even prevent the binding of unknown activator(s). To check if another member of the HSF family would replace HSF1, we analyzed HSF2 which can bind to the same DNA sequences as HSF1 [36]. HSF2 transcription was inhibited after heat shock, interestingly, much less in HSF1^def^ than in HSF1^prof^ MCF7 cells (see Appendix A). Surprisingly, HSF2 protein levels were significantly decreased after heat shock only in the absence of HSF1 (Figure 5D). ChIP-qPCR analyses showed that enhanced after heat shock binding of HSF1 in the sequences analyzed (*HSPH1*, *HSPD1*, as known HSF1-regulated genes, *ATF3*, *JUN* and *FOS* family members) correlated with decreased binding of HSF2 (except *FOSB*) in HSF1+ cells. HSF2 binding was generally weaker in untreated HSF1− cells than in HSF1+ cells and further decreased after heat shock (except for *HSPH1* and *HSPD1*) (Figure 5E). These results suggest that HSF2 did not compensate for the lack of HSF1.

### 3.5. Subsets of Heat Shock-Regulated Genes Are Expressed Differently in Human Cancers with Different Levels of HSF1

Proteotoxic stress leading to HSF1 activation is frequently observed in tumors (as well as HSF1 overexpression) [37,38]. To check whether and how the expression of genes regulated by heat shock in MCF7 cells may correlate with HSF1 expression levels in actual cancer tissue, we conducted differential expression tests between selected groups of cancer patients with high and low HSF1 levels (specified in Appendix A) based on RNA-seq data deposited in the TCGA database (Appendix A). Since the acquisition of DNA-binding competency depends on temperature and concentration of HSF1 [39], we assumed that HSF1 may be active in cancers with high levels of HSF1. We focused on genes selected as differentiating the heat shock response in MCF7 cells with different HSF1 levels, and, therefore, directly or indirectly dependent on HSF1 (subsets from Figure 1C and validation experiments). We expected that genes with a higher fold change in HSF1^prof^ cells than in HSF1^def^ cells would have higher expression levels in HSF1^high^ cancers while genes with a higher fold change in HSF1^def^ cells would have higher expression levels in HSF1^low^ cancers. In general, large fractions of these gene subsets met the above expectations but differences were observed between analyzed cancers (breast invasive carcinoma, BRCA, colon adenocarcinoma, COAD, and acute myeloid leukemia, LAML, which were selected as best suited to the MCF7, RKO, and HAP1 cell lines used in in vitro studies) (Figure 6A,B). In the case of the gene subset with a higher fold change in HSF1^prof^ cells, the signature was better preserved in ER+ than ER− breast cancers (Figure 6A). Especially, *HSPB8*, a known target of both HSF1 and ERα, but also other known HSF1 targets (*HSPD1*, *HSPA4L*) were not upregulated in ER−/HSF1^high^ breast cancers (Figure 6A; see also Appendix A showing the correlation analyses between *HSF1* and *HSPs* expression). This is in agreement with the recent report on the cooperation of HSF1 with the estrogen receptor [17], and on the other hand, indicates that MCF7 cells are a good model cell line for ER-positive breast cancer. Other typical HSF1 targets upregulated by heat shock (e.g., *HSPA1A*, *HSPA1B*, *HSP90AB1*) showed higher expression levels in all HSF1^high^ breast cancers. Interestingly, no one from these HSP genes was upregulated in HSF1^high^ COAD and LAML, although LAML very well reflected the signature of this subset of genes (when up/down ratio was compared; Figure 6A). Interestingly, a signature of genes showing stronger heat-inducibility in HSF1-deficient cells (77 from Figure 1C and Appendix A from the validation experiment) was better preserved than the signature of gene subset with a higher fold change in HSF1-proficient cells (Figure 6A,B). This signature was the worst preserved in LAML, which was especially evident when only selected genes potentially regulated by HSF1 (i.e., with detected binding of HSF1) were analyzed (Figure 6C), indicating a different HSF1 action in leukemia and adenocarcinoma. *ATF3*, *FOS*, and *FOSB*, i.e., targets of HSF1 (Figure 5B) highly induced in HSF1-deficient cells (Figure 5C), were also expressed at a higher level in HSF1^low^ BRCA, but only in ER+, not ER− (Figure 6B,C and Appendix A). On the other hand, *JUN*, *JUND*, and *FOSL1*, another set of potential targets of HSF1 upregulated after heat shock, were expressed at higher levels (or not significantly changed) in all analyzed HSF1^high^ cancers (Figure 6B,C). Interestingly, the strongest positive correlation was found between *HSF1* and *JUN* family members in LAML (Appendix A), while *HSPs* levels were not correlated with *HSF1* levels in this cancer (Appendix A). This suggests that the transcriptional regulatory network generated during heat shock and dependent on HSF1 may be differently regulated in the analyzed cancer types. Specifically, HSF1 may inhibit an accelerated transcriptional response after heat shock in ER+ BRCA but not in LAML.

## 4. Discussion

When we aimed to study the transcriptional response to heat shock in HSF1-deficient cells, we anticipated it to be generally suppressed. As expected, the majority of genes involved in the HSF1-mediated heat shock response were less activated in the absence of HSF1. Interestingly, however, we identified a set of genes that were more strongly activated in such cells. This set included heat shock-activated *JUN* and *FOS* family members, as well as *ATF3*, and other stress-activated transcription factors. Furthermore, HSF1 that was bound to the promoter/regulatory sequences of these genes (except *JUNB*) could play a dual (activatory/suppressory) role in their regulation. HSF1 binding after heat shock correlated with upregulation of the transcription (except *FOSL2*) suggesting an activatory HSF1 function. However, the transcription activation of these genes was not repressed in HSF1-deficient cells (*JUN*, *JUND*, *FOSL1*) or was even stronger in the case of *FOS*, *FOSB*, and *ATF3* suggesting an inhibitory HSF1 function. HSF1 was already found to upregulate *JUN* and *ATF3* expression [40,41] while the inhibitory action of HSF1 was proposed in the case of *FOS* activation by RAS [42]. On the other hand, ATF3 can directly activate *HSF1* transcription in response to increased cAMP levels (at least in thermogenic tissues, such as brown and beige fat) [43]. Hence, HSF1 and other stress-induced HSF1-regulated transcription factors could form a regulatory network that controls transcriptional response to proteotoxic stress. Observed by us, the dual function of HSF1 seems to be tissue-specific: TCGA data analysis indicates that such an HSF1-dependent regulatory network may not exist in ER-negative breast cancer or acute myeloid leukemia. In particular, the expression of *FOS*, *FOSB*, and *ATF3* did not correlate with the low HSF1 levels in these cancer types. We assume that the observed differences between cancers may result from various posttranslational modifications of HSF1, its protein partners, and chromatin organization in target genes. Thus, in addition to the fact that HSF1 drives a transcriptional program distinct from heat shock to support malignant phenotype [44], proteotoxic stress (e.g., induced by therapeutic hyperthermia) can elicit different effects depending on the level of HSF1 and the type of cancer. In addition to the different levels of chaperones that are dependent on HSF1, the inflammatory response mediated by AP1 and ATF3 (regulated by HSF1) can differ from tumor to tumor.

Enhanced activation of HSF1-regulated genes in the absence of HSF1 could result from the binding of another transcription activator(s) in sites normally occupied by HSF1. Heat shock elements (HSEs) are also recognized by the structurally related HSF2. However, HSF2 levels dropped dramatically after heat shock in HSF1-deficient MCF7 cells, and its binding to the promoters of *ATF3*, *JUN*, and *FOS* family members was weaker than in HSF1+ cells and was not increased after heat shock. Moreover, although HSF2 was shown to accompany HSF1 to upregulate the transcription [32,45], it did not compensate for the lack of HSF1 (e.g., in MEF cells; [15]). In addition to HSFs, STATs were identified as transcription factors that recognize sequences similar to HSE [46] and were found by the BETA tool as potential candidates for regulating this set of genes (not shown). Among STATs family members, STAT3 but not STAT1 may be activated after heat shock [47,48] which suggests that it should be tested for its ability to replace HSF1. Interactions with transcription factors that bind close to HSE (interestingly, AP1/FOS/NRF2 binding site was identified in 100 bp regions surrounding HSF1-binding peaks [44]) or are in a common regulatory loop are also possible. It was previously suggested that HSF1 may be involved in remodeling 3D chromatin architecture in response to temperature stress [49] and recruited into CTCF-occupied, non-transcribed chromatin [32]. We showed here the interaction of HSF1 with CTCF, a known regulator of chromatin architecture that was proposed to act in repressing specific HSF1 target genes [33]. This indicates that HSF1 may regulate transcription directly (by binding to the promoters) and indirectly by participating in chromatin organization. Both proteins are expressed ubiquitously, so this may be a universal mechanism not related to the cell type. HSF1 is also able to participate in chromatin organization by interacting with the estrogen receptor α (ERα) [17], thus the absence of HSF1 may result in changes in the organization of chromatin loops affecting transcription. Interactions of HSF1 with ERα are cell type-specific and could account for the observed differences between ER-positive and ER-negative tumors in the expression of HSF1-dependent genes.

Among pathways associated with genes more strongly activated by the heat shock in HSF1-deficient cells, the most noteworthy was TNFα signaling via NFκB. ATF3 and JUN&FOS family members, directly regulated by HSF1, are important components of this pathway. Further studies confirmed that the HSF1 deficiency resulted in higher TNFα production and release after heat shock, while it was rather suppressed in HSF1+ cells. Similarly, overproduction of TNFα was noted in HSF1 null mice during endotoxemia [50,51]. TNFα has both autocrine and paracrine functions that amplify or shape the signaling via NFκB (also stimulating AP1 transcriptional activity [52] or inducing ATF3 [53]), thereby promoting an inflammatory response [54]. It was shown previously that NFκB signaling can not be fully activated by TNFα for several hours after heat shock [55], and tends to recover faster in cells with normal HSF1 levels compared to cells deficient in HSF1 [56]. Nevertheless, we observed the highest production of TNFα 6-24 h post-treatment, when NFκB signaling was already restored. Thus, we assume that TNFα production observed after heat shock, although low, may have physiological consequences: it was shown that single cells were able to respond to picomolar TNFα (3 pg/mL) [57]. Our results suggest that inflammatory signals induced by proteotoxic stress may be further self-amplified via TNFα and NFκB only in cells with low expression of HSF1. Additionally, we observed HSF1-dependent activation of some genes that, in our opinion, may be involved in the inhibition of NFκB signaling after heat shock, e.g., *NKRF* (NFKB repressing factor), *NFKBID* (NFKB inhibitor delta), *TRAF2* (TNF receptor associated factor 2), and *TRAF3* (TNF receptor associated factor 3). Taken together, our results indicate that HSF1 can inhibit an accelerated transcriptional response (and, therefore, inhibit the excessive synthesis of inflammatory cytokines) which, together with the upregulation of chaperone proteins, may prevent inflammation but this action may be tissue-specific.

## Figures and Tables

**Figure 1 cells-11-02510-f001:**
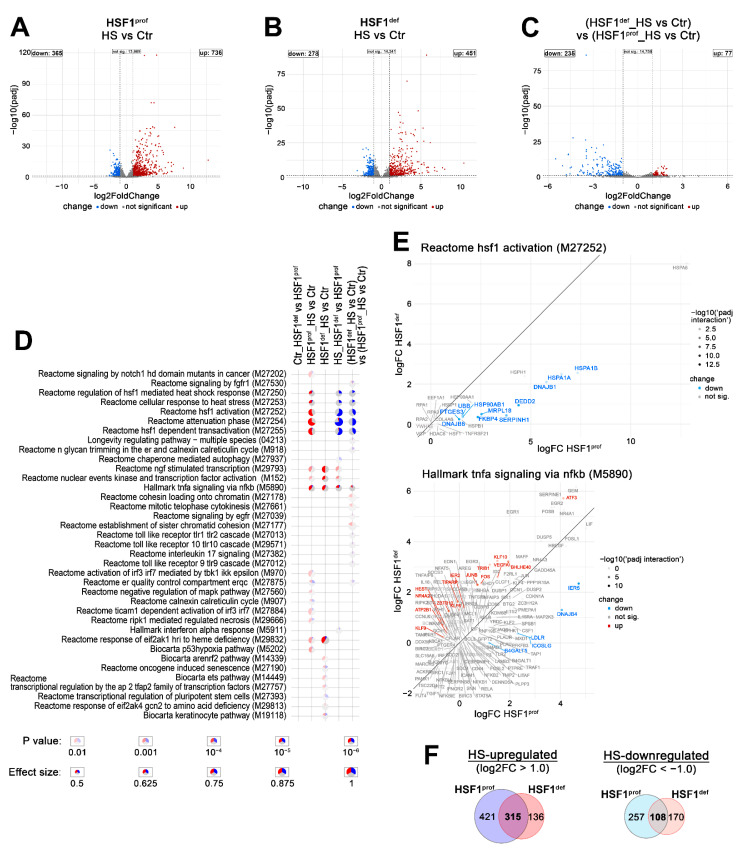
HSF1 deficiency affects the transcriptional response to heat shock in MCF7 cells. (**A**–**C**) Volcano plots of RNA-seq results showing the differentially expressed genes in response to heat shock (HS) in cells with normal (**A**) and reduced (**B**) levels of HSF1 (HSF1^prof^ and HSF1^def^, respectively), and a comparison of the response in both cell variants: changes induced by heat shock in HSF1-deficient cells were compared to changes induced in HSF1-proficient cells (**C**). Plots with gene labels are shown in Appendix A. (**D**) Geneset enrichment analysis showing significant terms from the Hallmark, Reactome, BioCarta, Pid, and KEGG gene sets collection detected in heat-shocked HSF1^prof^ and HSF1^def^ cells, as well as differences between cell variants. Blue—a fraction of downregulated genes, red—a fraction of upregulated genes. (**E**) Scatterplots of log2-fold-changes of genes associated with selected genesets upon HS stimulation in HSF1^prof^ (on *X*-axis) and HSF1^def^ (*Y*-axis) cells. (**F**) Overlap of genes stimulated or repressed after heat shock in both cell line variants.

**Figure 2 cells-11-02510-f002:**
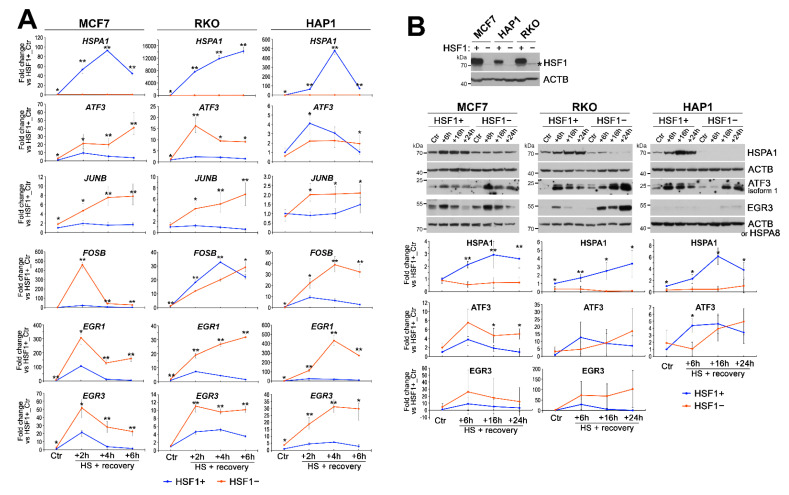
Regulators of cellular stress and growth response are differently expressed in heat-shocked HSF1+ and HSF1− cells. (**A**) Expression of *HSPA1*, *ATF3*, *JUNB*, *FOSB*, *EGR1*, and *EGR3* analyzed by RT-qPCR in MCF7, HAP1, and RKO cells exposed to elevated temperature (HS: 43 °C/1 h + recovery 37 °C/2 h, 4 h, or 6 h) in relation (fold change) to untreated control (Ctr) in HSF1+ cells. In the case of MCF7, a different cell model was used than for RNA-seq. Statistically significant differences between HSF1+ and HSF1− in each time point: ** *p* < 0.001, * *p* < 0.05. (**B**) Western blot analyses of HSF1 (the asterisk shows non-specific bands) in untreated HSF1+ and HSF1− MCF7, HAP1, and RKO cells and HSPA1, ATF3, and EGR3 up to 24 h of recovery from one-hour heat shock. ACTB or HSPA8 were used as loading controls. Results of densitometric analyses (n = 2–4) are shown below blots (note that highly variable EGR3 expression resulted in large standard deviations). Statistically significant differences between HSF1+ and HSF1− in each time point: ** *p* < 0.001, * *p* < 0.05.

**Figure 3 cells-11-02510-f003:**
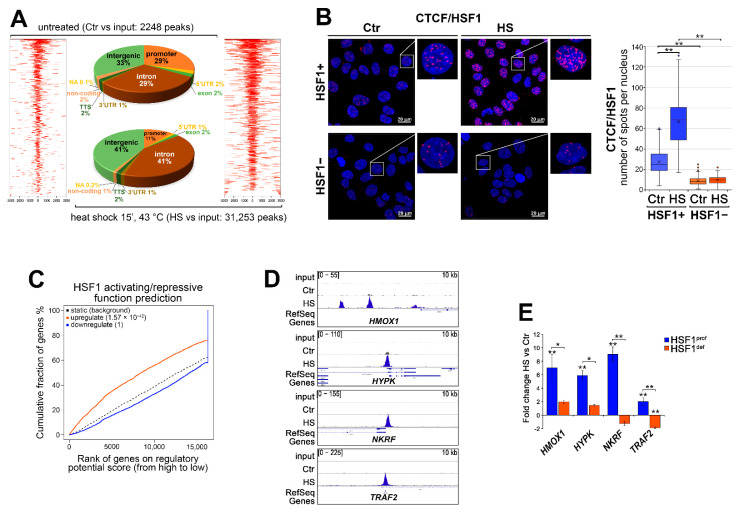
HSF1 binding across the genome is remodeled after heat shock in MCF7 cells possibly involving CTCF. (**A**) Heatmaps and distribution of HSF1 binding sites within different genomic regions in untreated and heat-shocked cells (based on ChIP-seq data). Heatmaps depict all HSF1 binding events centered on the peak region within a 3 kb window around the peak. Peaks in each sample were ranked on intensity. (**B**) Interactions between CTCF and HSF1 assessed by PLA (red spots) in MCF7 cells (HSF1− cells serve as a negative control), untreated and after HS treatment (15 min at 43 °C). DNA was stained with DAPI. Scale bar, 20 μm. Representative nuclei are enlarged. The number of spots per nucleus is shown in boxplots (which represent the median and the mean, upper and lower quartiles, maximum and minimum, and outliers). ** *p* < 0.001. (**C**) Prediction of activating/repressive function of HSF1 based on ChIP-seq and RNA-seq data integration using the BETA tool. The red, blue, and dashed lines represent the upregulated, downregulated, and nondifferentially expressed genes (as background), respectively. (**D**) Examples of HSF1 peaks identified in ChIP-seq analyses and visualized by the IGV browser in untreated cells (Ctr) and after heat shock (HS). The scale for each sample is displayed in the left corner, length of the region shown—in the right corner. (**E**) Changes in expression of genes activated after heat shock in HSF1^prof^ but not in HSF1^def^ cells (extracted from RNA-seq data). ** *p* < 0.001, * *p* < 0.05 (significance of differences versus the corresponding control or between cell variants).

**Figure 4 cells-11-02510-f004:**
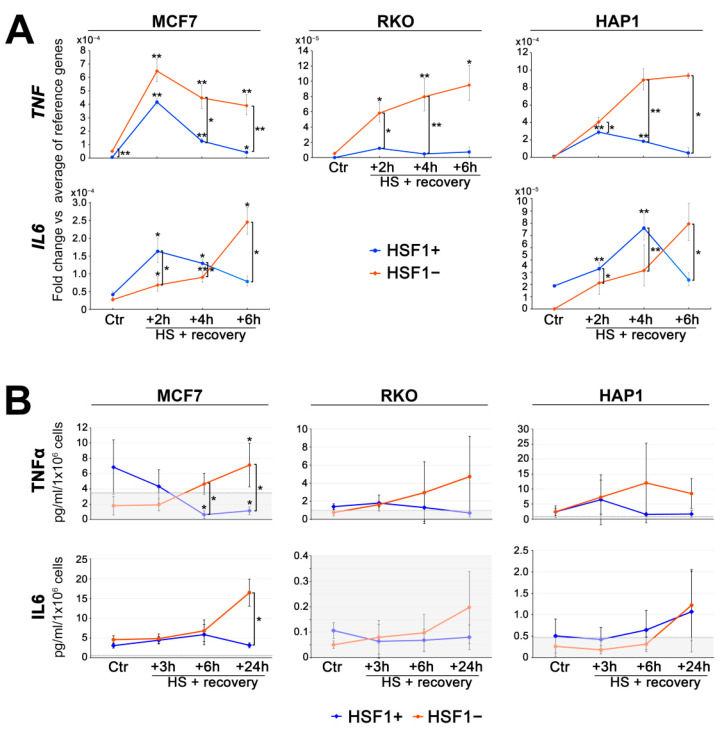
The production of inflammatory cytokines in HSF1+ and HSF1− MCF7, HAP1, and RKO cells cells. (**A**) Transcriptional changes of TNF and IL6 in response to heat shock. Expression was analyzed by RT-qPCR in cells exposed to elevated temperature (HS: 43 °C/1 h + recovery 37 °C/2 h, 4 h, or 6 h), and fold changes were calculated versus the average of reference genes (due to the undetectable transcript level in some untreated control, Ctr, samples). IL6 was not detected in RKO cells. (**B**) The level of TNFα and IL6 released by cells exposed to elevated temperature (HS: 43 °C/1 h + recovery 37 °C/3 h, 6 h, or 24 h) was analyzed by ELISA. Values in the gray box are below the assay quantitation range. The difference between treated and untreated cells, as well as between HSF1+ and HSF1− in each time point: ** *p* < 0.001, * *p* < 0.05.

**Figure 5 cells-11-02510-f005:**
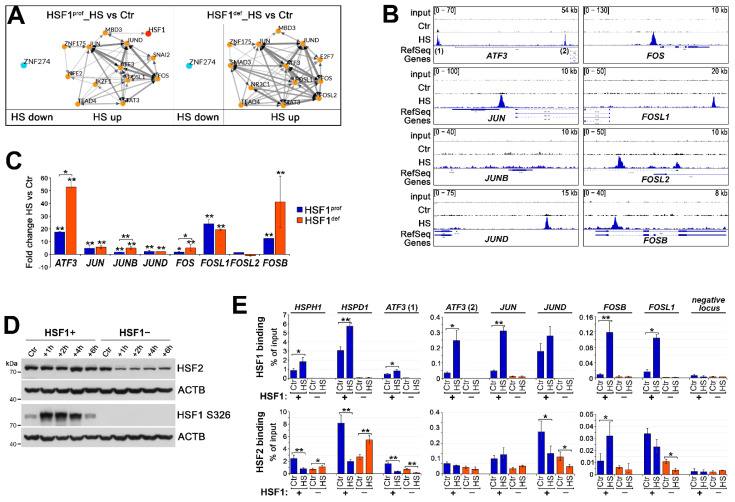
Heat shock initiates the formation of a transcriptional regulatory network that may be stronger in HSF1-deficient MCF7 cells. (**A**) Networks of upstream regulators of HS-stimulated (HS up) or repressed (HS down) genes (identified by RNA-seq analyses in HSF1^prof^ and HSF1^def^ cells) predicted using the ChEA3_ReMap (FDR < 0.05, top 13 factors are shown). (**B**) HSF1 peaks in the regulatory regions of *ATF3* and *JUN* and *FOS* family members identified in ChIP-seq analyses and visualized by the IGV browser in untreated cells (Ctr) and after heat shock (HS). The scale for each sample is displayed in the left corner, length of the region shown—in the right corner. (**C**) Heat-induced changes in expression of selected genes in HSF1^prof^ and HSF1^def^ cells (extracted from RNA-seq data). ** *p* < 0.001, * *p* < 0.05 (significance of differences versus the corresponding control or between cell variants). (**D**) Western blot analysis of HSF2 levels up to 6 h of recovery from heat shock in HSF1+ and HSF1− cells. ACTB was used as loading control, and HSF1 S326 phosphorylation was analyzed to monitor HSF1 activation after heat shock. (**E**) HSF1 (upper panel) and HSF2 (bottom panel) binding (by ChIP-qPCR; % of input) in selected sequences in untreated (Ctr) and after HS treatment (43 °C, 15 min) HSF1+ and HSF1− cells. Two analyzed binding sites in *ATF3* are depicted in (**B**). ** *p* < 0.001, * *p* < 0.05.

**Figure 6 cells-11-02510-f006:**
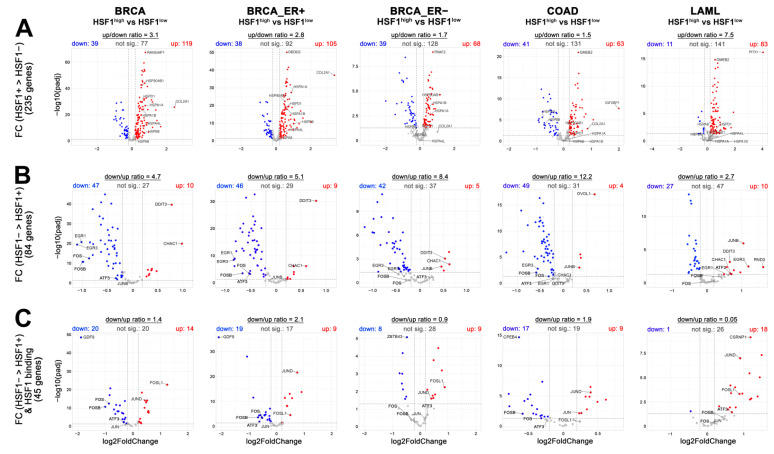
Differential expression analysis in human cancers with different levels of HSF1. Volcano plots comparing the expression of selected genes in BRCA (breast invasive carcinoma; ER+ and ER−, estrogen receptor-positive and negative, respectively), COAD (colon adenocarcinoma), and LAML (acute myeloid leukemia) cases with high and low *HSF1* levels. Red/blue dots show genes with higher/lower expression in HSF1^high^ than in HSF1^low^ cancer cases. Signature of genes selected as differentiating the heat shock response in HSF1-proficient and HSF1-deficient MCF7 cells (shown in Figure 1C): (**A**) with higher fold change (FC) in HSF1^prof^ cells (expected to be up in volcano plots); (**B**) with higher fold change in HSF1^def^ cells (expected to be down in volcano plots). (**C**) Signature of HSF1 targets with higher fold change in HSF1^def^ cells.

**Table 1 cells-11-02510-t001:** List of genes potentially upregulated by HSF1 after heat shock (genes with a lower rank product calculated by the BETA tool are more likely to be HSF1 target genes) in HSF1^prof^ MCF7 cells that showed stronger upregulation (at least 2-fold as assessed by RNA-seq analyses) in HSF1^def^ cells. RT-qPCR analysis was performed on another the HSF1+ and HSF1− MCF7 cell model.

	Rank Product	Gene Symbol	Fold Change (FC) ^1^		Difference (FC HSF1^def^/HSF1 ^prof^)	Difference (padj)	Difference Confirmed by RT-qPCR
			HSF1^prof^	HSF1^def^			
1.	0.00818	LINC00324	2.87	10.98	3.82	0.030	+
2.	0.00749	TRIB1 ^2^	2.30	8.54	3.71	0.020	+
3.	0.00060	FOSB ^2^	12.47	41.14	3.30	0.173	+
4.	0.00410	EGR2 ^2^	13.37	41.43	3.10	0.567	na
5.	0.00005	ATF3 ^2^	17.40	52.79	3.03	0.003	+
6.	0.01065	DUSP10	1.70	5.12	3.02	0.0056	+
7.	0.00229	RRAD	480.7	1413.7	2.94	0.845	na
8.	0.00134	GEM	24.72	72.56	2.94	0.362	na
9.	0.004546	FOS ^2^	1.82	5.12	2.82	0.023	na
10.	0.003438	ERRFI1 ^2^	1.97	5.01	2.54	3.03 × 10^−7^	na
11.	0.01135	CCDC89	4.54	10.97	2.41	0.603	na
12.	0.00363	CSRNP1 ^2^	4.25	9.35	2.20	0.153	na
13.	0.00023	NR4A1 ^2^	17.61	38.63	2.19	0.375	na
14.	0.01285	ID2 ^2^	2.44	5.35	2.19	0.348	+

^1^ all heat-induced fold changes are significant. ^2^ regulators of gene expression (GO:0010468). na, not analyzed.

## Data Availability

The data presented in this study are openly available in the NCBI GEO database: acc. no. GSE159802, acc. no. GSE137558 (GSM4081758, GSM4081759, and GSM4081762).

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
