# Peer review of "HSF1 Can Prevent Inflammation following Heat Shock by Inhibiting the Excessive Activation of the ATF3 and JUN&FOS Genes"

_cells, 2022, doi:10.3390/cells11162510_

Round 1

Reviewer 1 Report

In the manuscript entitled “Heat shock-activated transcriptional regulatory network and inflammatory responses may be more robust and persist longer in HSF1-deficient cancer cells”, the authors study the different abilities of cancer cells expressing or not HSF1 to regulates the heat shock response in terms of gene expression.

The study is interesting and the manuscript well written.

Minor revisions:

-Similar approaches have been used in other studies (i.e. Mendillo et al., 2012 PMID: 22863008) and connections between HSF1 and inflammatory responses have been already highlighted (Xiao et al,1999 PMID: 10545106; Chen et al., PMID: 16061216, PMID: 22753951, PMID: 33381262; etc). The results must be discussed considering previous data, highlighting the novelty of this research.

-Paragraph 3.2: a clearer explanation of the data involving CTCF in HSF1 activity would help comprehension. Pictures are clear, but text should be improved.

-Discussion: “It was shown previously that NFκB signaling cannot be fully activated by cytokines for several hours after heat shock [47], and tends to recover faster in cells with normal HSF1 levels [48]”.

Could you better explain and discuss this point? It recovers faster in respect to HSF1 null? Can you better discuss your results in respect to this data and on heat-induced NF-KB signaling impairment?

-Please show a Western blot analysis for HSF1 for wild type and KO cell lines.

-Figure 1c: it is not completely clear what type of comparative analysis has been performed.

- Page 2, line 62: “Dysregulation of HSF1 and its target genes are associated with the disease.” What disease are you referring to?

-Page 15, lane 534 “Observed by us dual function of HSF1 seems to be tissue-specific.” This point deserves a more accurate discussion.

-Figure panels and supplementary tables are not cited in alphabetical or numerical order.

-I suggest changing the title to a shorter and more direct one.

Reviewer 2 Report

The authors investigated various consequences of HSF1 deficiency in breast adenocarcinoma cells using various methods. The obtained results are of interest and definitely should be published. Specifically, it was shown that several genes and signal pathways (ATF3, FOS, JUN etc) were activated in the cell line with HSF1 deficiency. However, there are a few queries that should be addressed. 

1. It is not clear how efficient HS treatment was in these experiments i.e. to what extent the synthesis of all Hsps was inhibited. It will be also nice to indicate in M@M the duration of the HS treatment.

2. There are at least three lines of HSF1-null mice that were studied in detail. It is necessary at least briefly to discuss these data.

3. The are data demonstrating that paradoxically, HSF1-null mice are more resistant to oncogenesis. It will be important to discuss these data taking into account the results accumulated by the authors.

4. Although the authors performed Western blotting using various a-bodies, 2D-gel experiments using protein labeling under normal conditions and after HS (35S-methionine) will definitely provide a lot of valueble information concerning the pattern of protein synthesis under normal conditions and after HS treatment.

Reviewer 3 Report

Janus et al., nicely compared the network of genes regulated by heat shock in cancer cells in presence or in absence of HSF1, the master regulatory protein of heat shock and other stresses. They used computational analyses in MCF7 breast adenocarcinoma cells modified for HSF1 expression level and that undergo heat shock.  They identified a set of genes that was stronger and longer induced in HSF1-deficient cells than in cells with normal level of HSF1. Those genes included ATF3 and some AP-1 transcription factors. They also found that inflammatory responses last longer in HSF1-deficient cells. In addition, they replicated the experiments in the colorectal cancer cell line RKO and in the chronic myelogenous leukemia cell line HAP1. The experiments were conducted with the correct experimental design, and the results obtained well support the advanced hypotheses. Experimental materials and methodologies are well described. Below are my comments regarding the study.

Major points:

1) Although the cell models that are used in this study have been previously described by the authors in other published articles, and although the authors showed HSPA1 expression, a canonical target of HSF1, to confirm the functional knockout of HSF1 in their cell models, the authors need to provide data on the down-regulation of HSF1 expression specifically in the same cells that were used in this study, namely HSF1+ and HSF1− MCF7, RKO, and HAP1 cells.

2) I have some concerns about the interaction with CTCF: The authors need to explain why they looked at the interaction between HSF1 and CTCF. Is CTCF a HSF1 target gene? Does the CTCF level of expression increase upon heat shock? What is the level of CTCF in HSF1- cells? What is the effect of CTCF on HSF1 transcriptional activity? The authors claim that HSF1 has rather an activating function, not repressive. Is this because of CTCF? How CTCF could remodel the bonding of HSF1 after HS?  The authors could perform a luciferase reporter assay with a Heat Shock Element (HSE/HSF1) luciferase Reporter in presence of CTCF. The authors need also to quantify the expression of HSF1 target genes such as HMOX1, HYPK, NKRF, TRAF2 in HSF1+ cells upon CTCF overexpression or downregulation by siRNA and after HS.

3) Links exist between pro-inflammatory cytokines, such as TNF-α and IL6, and the AP-1 transcription factors, but also with ATF3 transcription factor. Indeed, TNF-α stimulate AP-1 transcriptional activity and can also induce ATF3, and on another hand, AP-1 can induce IL6 expression. Authors should discuss these interactions in the context of their data and cite relevant literature. 

Minor points:

1) The authors need to explain how they estimate the total number of viable and dead cells when they measured the secreted cytokines by ELISA in the Materials and Methods.

2) The authors need to quantify the proximity ligation assay.

3) The authors could justify the choice of ATF3 and EGR3 for validation at the protein level.

Round 2
